# Simultaneous Determination of One-Carbon Folate Metabolites and One-Carbon-Related Amino Acids in Biological Samples Using a UHPLC–MS/MS Method

**DOI:** 10.3390/ijms25063458

**Published:** 2024-03-19

**Authors:** Yi Ling, Mei Tan, Xiaoyun Wang, Ziyi Meng, Xiaodong Quan, Hosahalli Ramaswamy, Chao Wang

**Affiliations:** 1Department of Food Science and Technology, Jinan University, Guangzhou 510632, Chinatanmei@stu2020.jnu.edu.cn (M.T.); wangxy@stu2022.jnu.edu.cn (X.W.); mengzy@stu2020.jnu.edu.cn (Z.M.); 15625145228@163.com (X.Q.); 2Department of Food Science and Agricultural Chemistry, Macdonald Campus of McGill University, Montréal, QC H3A 0G4, Canada

**Keywords:** folate, biological samples, amino acids, UHPLC–MS/MS

## Abstract

One-carbon folate metabolites and one-carbon-related amino acids play an important role in human physiology, and their detection in biological samples is essential. However, poor stability as well as low concentrations and occurrence in different species in various biological samples make their quantification very challenging. The aim of this study was to develop a simple, fast, and sensitive ultra-high-performance liquid chromatography MS/MS (UHPLC–MS/MS) method for the simultaneous quantification of various one-carbon folate metabolites (folic acid (FA), tetrahydrofolic acid (THF), *p*-aminobenzoyl-L-glutamic acid (pABG), 5-formyltetrahydrofolic acid (5-CHOTHF), 5-methyltetrahydrofolic acid (5-CH_3_THF), 10-formylfolic acid (10-CHOFA), 5,10-methenyl-5,6,7,8-tetrahydrofolic acid (5,10-CH^+^-THF), and 4-α-hydroxy-5-methyltetrahydrofolate (hmTHF)) and one-carbon-related amino acids (homocysteine (Hcy), methionine (Met), *S*-ade-L-homocysteine (SAH), and *S*-ade-L-methionine (SAM)). The method was standardized and validated by determining the selectivity, carryover, limits of detection, limits of quantitation, linearity, precision, accuracy, recovery, and matrix effects. The extraction methods were optimized with respect to several factors: protease–amylase treatment on embryos, deconjugation time, methanol precipitation, and proteins’ isoelectric point precipitation on the folate recovery. Ten one-carbon folate metabolites and four one-carbon-related amino acids were detected using the UHPLC–MS/MS technique in various biological samples. The measured values of folate in human plasma, serum, and whole blood (WB) lay within the concentration range for normal donors. The contents of each analyte in mouse plasma were as follows: pABG (864.0 nmol/L), 5-CH_3_THF (202.2 nmol/L), hmTHF (122.2 nmol/L), Met (8.63 μmol/L), and SAH (0.06 μmol/L). The concentration of each analyte in mouse embryos were as follows: SAM (1.09 μg/g), SAH (0.13 μg/g), Met (16.5 μg/g), 5,10-CH^+^THF (74.3 ng/g), pABG (20.6 ng/g), and 5-CH_3_THF (185.4 ng/g). A simple and rapid sample preparation and UHPLC–MS/MS method was developed and validated for the simultaneous determination of the one-carbon-related folate metabolites and one-carbon-related amino acids in different biological samples.

## 1. Introduction

Folates, also known as vitamin B9, act as co-substrates and serve as regulatory molecules. In several key enzymatic reactions, folate carries one-carbon residues and acts as a mobile cofactor to participate in one-carbon metabolism for biosynthetic reactions, including de novo purine and thymidylate synthesis and the methylation of homocysteine to methionine [1,2]. 

However, folate deficiency is widespread and occurs worldwide, both in developing and developed countries [3]. It is considered as one of the most common nutritional deficiencies which can lead to a variety of disorders including neural tube defects, megaloblastic or macrocytic anemia, occlusive vascular disease, and others [4,5]. On the other hand, excessive folate intake has also been associated with adverse health effects, such as autism, negative neurocognitive development, and some cancers [6].

Folates comprise three parts, including *p*-aminobenzoate, a pteridine moiety, and different numbers of glutamate moieties (Figure 1). The backbone of the folate structure can be subjected to a variety of transformations such as reductions, methylations, and formylations, and the term “folates” actually encompasses all of these modifications [7]. However, humans are not able to synthesize folate de novo and have to rely on food sources or supplements. In addition to being influenced by folate intake, folate levels in humans are influenced by genetic polymorphisms, homocysteine (Hcy) metabolism, and a variety of environmental factors, such as alcohol, smoking, and some medicines that interfere with folate metabolism, such as trimethoprim and methotrexate [8].

Folate levels in different biological samples such as serum, plasma, and WB provide different information at the clinical level. The serum folate concentration is considered an indicator of recent folate intake, but there is no established cut-off serum concentration of folate for determining the risk of neural tube defects [9]. On the other hand, the folate concentrations in WB may reflect the long-term folate status with a biological half-life of 8 weeks; therefore, this parameter is generally used to predict the risk of neural tube defects [10]. Finally, folate levels in plasma are well-known biomarkers of disease and may reflect dietary folate intake [11]. Furthermore, in early studies involving pregnant mice with neural tube defects, an alteration in the folate pattern of the embryo was observed; therefore, this aspect also merits investigation [12]. 

The extraction of endogenous folate from various biological samples typically employs two distinct methodologies. The first method involves the extraction of biological samples, such as serum and plasma, without the hydrolysis of polyglutamated folates into monoglutamated folates. Conversely, the second method necessitates the hydrolysis of polyglutamated folates into monoglutamated folates by endogenous or exogenous folylpolyglutamate hydrolases when analyzing samples like whole blood (WB) and embryos. Additionally, different folate species are susceptible to degradation or conversion under varying conditions of light exposure, oxygen presence, specific temperatures, and specific pH levels, thereby rendering the analysis quite challenging. To enhance folate stability, it is imperative to expeditiously extract the samples and incorporate antioxidants, such as ascorbic acid (AA) and 2-mercaptoethanol (2-MCE), during the sample preparation process [13].

Another challenge associated with the quantification of endogenous folate in different biological samples is its presence in extremely low concentrations. For decades, various analytical methods have been developed such as microbiological assay, radioimmunoassay, capillary electrophoresis, and UHPLC–MS/MS [13,14,15,16]. Microbiological assay uses *Lactobacillus casei* as the test organism which requires folate to proliferate. Microbiological assay has several disadvantages: it is difficult and time-consuming, has poor precision, and has relatively low specificity (unable to distinguish different folate species) [17]. The radioisotope immunoassay and biosensor approaches, which are expensive for routine analysis and also cannot differentiate folate species. The shortcomings of the above methods could be overcome by UHPLC–MS/MS methods which are rapid, highly sensitive, and selective and could be effectively applied for the quantification of folate metabolites in biological samples [18].

Currently, total folate is often used to clinically interpret folate status; however, the levels of specific folate vitamers and related metabolites in different biological samples would provide more useful information. In folate metabolism, there are five one-carbon substituted folate derivatives in vivo, and these derivatives carry one-carbon units ranging from formate to methanol and serve unique one-carbon transfer reactions. The predominant folate in blood is 5-CH_3_THF, and it delivers its own methyl groups to Hcy to form Met and THF [5,17]. Furthermore, some of the Met is then transformed to form SAM under catalysis by Met adenosyltransferase. A low-folate status could result in the accumulation of Hcy, which further results in the accumulation of SAH [19]. In addition, other folate vitamers can be converted to 5-CH_3_THF as shown in Figure 2. For example, 5,10-CH_2_THF could be converted into 5-CH_3_THF under the action of the enzyme of 5,10-CH_2_THF reductase (MTHFR). This reaction is very important in the maintenance of appropriate levels of 5-CH_3_THF when the folate supply is limited [20]. A storage form of the oxidation state of formate, 5-CHOTHF, can interconvert with 5,10-CH^+^THF [21], and 5-CHOTHF is not used as a cofactor for folate-dependent biosynthetic reactions; instead, it is thought to be an intracellular storage form of folate in dormant cells [22]. In addition, folate catabolism is a major route of folate turnover in humans and involves cleavage of the C9-N10 bond, producing a pterin (C_6_H_5_N_5_O) and pABG. In addition, 5-CH_3_THF can be oxidized to 5-CH_3_-5, 6-dihydrofolate under mild oxidation conditions (oxygen). Under more severe conditions (hydrogen peroxide), 5-CH_3_THF is oxidized further to a compound originally identified as hmTHF [23]. Re-examination of the identity of this compound has suggested that it has a pyrazino-s-triazine structure.

Therefore, the objective of this investigation is to determine one-carbon folate metabolites and amino acids, namely, FA, THF, pABG, 5-CHOTHF, 5-CH_3_THF, 10-CHOFA, 5,10-CH^+^THF, hmTHF, Hcy, Met, SAH, and SAM, in different biological samples (e.g., mouse serum, plasma, WB, and embryo) using the ultra-high-performance liquid chromatography–mass spectrometry/mass spectrometry (UHPLC–MS/MS) technique. The UHPLC–MS/MS methodology offers a streamlined and expeditious sample preparation process and is characterized by a brief analytical run-time. Upon standardization and validation, this method can be leveraged for high-throughput routine analysis, particularly in the realms of bioavailability and epidemiological research.

## 2. Results

### 2.1. Method Optimization

As the methods for extracting folate from plasma and serum have been extensively evaluated in previous studies [24,25,26], these were not optimized in this study. However, the method has not been well-optimized for the determination of folate in embryos and WB. Because native folate can be trapped in the embryo matrix, for example, in proteins and starch, solutions of a protease–amylase mixture were added, and the results are shown in Table 1. By using the UHPLC–MS/MS determination, the major folate species in these embryos were 5,10-CH^+^THF, 5-CH_3_THF, and folate oxidation product pABG, and their quantitation results are shown in Table 1 with 5,10-CH^+^THF having the highest content (58.8%), followed by 5-CH_3_THF (29.4%). The major amino acids included were Met, SAH, and SAM, with Met having the highest content (80.7%) followed by SAH (16.7%) and SAM (2.57%).

Considering the hydrolysis by protease–amylase, this treatment caused a significant loss of 5,10-CH^+^THF and 5-CH_3_THF, whose concentrations were no longer detectable (<LOD). However, it also caused 50% and 3.98-fold increases in pABG and Met, respectively. In addition, after the di-enzyme treatment, the residual content of SAH was very low (1.23%), and SAM content was decreased to below the LOD. 

Because the folates were present as polyglutamyl folates and were not accessible to GGH, the release of polyglutamyl folates from the matrix and deconjugation of polyglutamyl folate were deemed to be necessary. To deconjugate the polyglutamyl folate, the samples were incubated with folate-stripped rat serum, which provided a reliable source of the necessary GGH enzyme. Deconjugation times of 30, 60, and 90 min for the embryos were also investigated here, and the results are shown in Table 1. As the incubation time increased, there were no significant differences in 5,10-CH^+^THF, pABG, Met, and SAM (*p* > 0.05). After incubation for 90 min, 5-CH_3_THF and SAH were decreased by 80% and 55%, respectively. 

In addition, protein precipitation should be necessary for the extraction of folate from plasma, serum, and WB because folates are bound with folate-binding proteins (FBP). For serum and plasma, methanol was added to precipitate the proteins during the extraction [26,27]. However, the protein precipitation methods have never been optimized for WB. Two approaches to compare proteins precipitation were compared: methanol precipitation (method 1) and proteins’ isoelectric precipitation (method 2). The results are summarized in Table 2. The distribution of folate metabolites measured using these two methods was basically the same. Folate in the WB consisted of 5-CH_3_THF, 10-CHOFA, FA, hmTHF, and pABG. Of these, FA had the highest concentration (41.2–41.9%), and the major amino acids included Hcy, SAH, and Met, with Met being the predominant form (94.5–96.5%) followed by SAH (3.11–4.72%). However, the folate metabolite concentrations obtained using method 1 were significantly different from the concentrations obtained within method 2 (*p* < 0.05). The concentrations of all forms of folate metabolites were significantly higher when using method 2 compared with method 1, with the exception of hmTHF and Hcy. Compared to method 1, method 2 extracted more than 80%, 66.7%, 300%, 85.7%, 33.3%, and 106.7% of pABG, 5-CH_3_THF, 10-CHOFA, FA, SAH, and Met, respectively.

### 2.2. Method Validation

Mass spectrometry ion source conditions were optimized to achieve the highest possible responses for all folate species and “internal standards”. The MRM transitions were obtained using Mass Hunter Optimizer 12.1 method development software. The most abundant fragment ion was chosen as the quantifier, and the second most abundant was chosen as the qualifier. For “internal standards”, only one mass transition was used in order to increase the sensitivity of the method to analytes (Table 3). Analytes of folate species and related amino acids were detected in positive ion mode and were well-separated on the reversed phase column. Most of the molecules exhibited an adequately separated chromatographic peak that was easily distinguished from the baseline, as shown in Figure 3. Quantitation was performed based on the peak areas of the analytes and their corresponding “internal standards” using a linear regression model, whereas the peak identity was confirmed based on agreement with the retention time of the standards and MRM transitions, with the exception of hmTHF.

The method’s selectivity was assessed by analyzing a folate-stripped blank (rat serum, rat plasma, and WB). The selectivity met the criterion because no interfering peaks were observed at the same retention times for target analytes. In the determination of folates and amino acids, there was no interference in the stripped matrix and no obvious cross-contamination existed, as demonstrated upon injecting a 100 ng/mL standard solution.

The linearity of the method was checked by plotting the peak area versus the concentration over a wide concentration range. The slope and corresponding determination coefficient (R^2^) are given in Table 4. All the calibration curves are presented (0.992 ≤ R^2^ ≤ 0.999) and were suitable for the analysis of the samples. 

LOQs were set as the lowest calibration concentration showing an acceptable accuracy (100 ± 20%), precision (RSD ≤ 20%), and signal-to-noise ratio (≥10). The results for the LOD and LOQ are shown in Table 4. 

Accuracy and precision: The results obtained for the intra- and inter-day accuracy and precision of the method with low, medium, and high concentrations are presented in Table 5, and they are within acceptable limits. For accuracy, the intra-day and inter-day ranges were 92.17 to 112.53% and 92.10 to 115.29%, respectively. For precision, the intra-day and inter-day ranges were 1.01 to 7.55% and 0.69 and 12.04%, respectively.

The matrix effects and recoveries were investigated for mouse serum, mouse plasma, WB, and embryos. The results showed that the matrix effects lay between 89.98% and 114.36% for mouse plasma, 90.37% and 110.24% for mouse serum, 90.59% and 109.67% for WB, and 92.03% and 115.03% for embryos. The recoveries lay between 86.67% and 121.47% for mouse plasma, 88.67% and 108.13% for mouse serum, 84.39% and 113.44% for WB, and 85.76% and 119.53% for embryos. Although the values for the recoveries and matrix effects exceeded the 15% acceptance criterion, none exceeded 20% with the exception of the recovery of SAH in mouse serum. These results indicate that the UHPLC–MS/MS method developed in this study has little influence of the matrix effect, while the recoveries were all acceptable and the method is reliable and feasible.

### 2.3. Application of the Method

#### Distribution of Folate Species in Different Biological Samples

The practical utility of the validated method was tested. Human WB, plasma, and serum samples were processed and analyzed as described in the Section 4 Materials and Methods. For WB, the concentrations of each analyte were as follows: 5-CH_3_THF (54.1 nmol/L), 10-CHOFA (44.1 nmol/L), FA (131.1 nmol/L), pABG (92.7 nmol/L), Met (2.48 μmol/L), SAH (0.08 μmol/L), and Hcy (0.01 μmol/L). For plasma, the concentrations of each analyte were as follows: 5-CH_3_THF (13.6 nmol/L), hmTHF (24.6 nmol/L), pABG (1192.0 nmol/L), Met (20.5 μmol/L), SAH (0.04 μmol/L), SAM (0.38 μmol/L), and Hcy (0.64 μmol/L). For serum, the concentrations of each analyte were as follows: 5-CH_3_THF (38.8 nmol/L), hmTHF (12.3 nmol/L), pABG (74.0 nmol/L), Met (3.43 μmol/L), SAH (0.07 μmol/L), and Hcy (0.23 μmol/L). 

In mouse WB, there were more folate species including high contents of 5,10-CH^+^THF (123.4 nmol/L), 5-CH_3_THF (338.3 nmol/L), hmTHF (200.1 nmol/L), pABG (1050.1 nmol/L), and FA (35.3 nmol/L). For amino acids, the species included Met (196.2 μmol/L), SAH (1.55 μmol/L), and SAM (0.84 μmol/L). However, Hcy was not detected in mouse WB and plasma. For mouse serum, the detected folate species included 5-CH_3_THF (70.1 nmol/L), hmTHF (60.3 nmol/L), pABG (171.3 nmol/L), Met (1.88 μmol/L), SAH (0.03 μmol/L), and Hcy (0.15 μmol/L). The levels of Met in mouse WB were significantly higher than those in the other biological samples (Table 6). 

The practical utility of the new method was also tested on a large subset of mouse plasma (*n* = 80) and mouse embryos (*n* = 24). The contents of each analyte in mouse plasma were as follows: pABG (864.0 nmol/L), 5-CH_3_THF (202.2 nmol/L), hmTHF (122.2 nmol/L), Met (8.63 μmol/L), and SAH (0.06 μmol/L). The predominant natural folate form in mouse plasma was 5-CH_3_THF. The summary ranges for each analyte in mouse embryos were as follows: SAH (1.09 μg/g), SAM (0.13 μg/g), Met (16.5 μg/g), 5,10-CH^+^THF (74.3 ng/g), pABG (20.6 ng/g), and 5-CH_3_THF (185.4 ng/g).

There was a significant positive correlation between the concentrations of plasma folate catabolites (pABG) and the concentration of the oxidation product of 5-CH_3_THF (hmTHF) (R = 0.4775, *p* < 0.0001), as shown in Figure 4A. In addition, there was some positive correlation of the total concentration of pABG and hmTHF (R = 0.4918, *p* < 0.0001) with SAH, as shown in Figure 4B.

## 3. Discussion

The results of this study show that protease–amylase is not applicable for the extraction of folate and related amino acids from embryo samples. Protease–amylase treatment caused significant losses in 5,10-CH^+^THF and 5-CH_3_THF to below the LOD and caused 50% and 3.98-fold increases in pABG and Met concentrations. Folate is made up of pteridine, pABG, and glutamate, and pABG is the product of folate degradation. Therefore, the significant increase in pABG generated from 5,10-CH^+^THF and 5-CH_3_THF degradation during long-duration di-enzyme incubation poses complications. The unique increase in Met after di-enzyme treatment might be due to the hydrolysis of the enzyme, and similar results have been reported elsewhere [28,29].

It can be concluded that deconjugation for 30 min is the most suitable method for the extraction of embryonic folate, as long incubation times caused significant degradation of labile folates and amino acids. Minimizing the sample incubation time before UHPLC–MS/MS analysis made it possible to eliminate potential biases caused by losses in labile folate forms during this process [28].

For the extraction of folate and related amino acids from WB, this study presents a protein precipitation procedure that could release the folate from the binding proteins and decrease the matrix effect during UHPLC–MS/MS analysis. The discrepancy between methanol and proteins’ isoelectric precipitation can be explained by the fact that acidification to a pH below 4.5 diminishes the binding capacity of FBP more effectively than methanol, thus releasing folates [30]. In addition, methanol extraction takes a relatively long time, which can cause folate loss [28], as significant amounts of hmTHF were determined. For WB, these processes can indicate the long-term folate status with a biological half-life of 8 weeks and are usually used to predict the risk of neural tube defects. The accurate determination of folate concentrations in WB has great clinical significance, and therefore, reliable analytical methods are needed [31].

The measured folate values for human plasma, serum, and WB lay within the expected concentration ranges for normal donor folate values (5-CH_3_THF, 6–170 nmol/L) [10,32]. Wang et al. determined significant amounts of SAH (0.03 ± 0.013 μmol/L), SAM (0.12 ± 0.046 μmol/L), and Hcy (5.87 ± 1.69 μmol/L) in the serum of normal pregnant women [19]. Awwad et al. [33] analyzed serum folate concentrations and found that tHcy (17.1 μmol/L), SAH (0.02 μmol/L), and SAM (0.14 μmol/L) were present. Hannisdal et al. [23] analyzed folate and folate catabolites in human serum and found that pABG (0–13.7 nmol/L) and hmTHF (0–12.7 nmol/L) were present. In another study, Hannisdal et al. [34] also analyzed the folate degradation products and found pABG concentrations of 8–37 and 10–29 nmol/L in the human serum and plasma, respectively. 

Compared with undetectable levels of FA in human serum and plasma, there was a certain amount of FA present in human WB, but the level of Hcy in WB was lower than that in serum and plasma. This is probably because FA plays an important role as a cofactor in Hcy metabolism, promoting the remethylation of Hcy to Met [35]. Several genetic and nongenetic factors can cause hyperhomocysteinemia [2]. Low folate intake results in elevated Hcy concentrations in the plasma of healthy men, middle-aged adults, elderly populations, and postmenopausal women. Folate depletion accompanied by elevated Hcy concentrations is associated with the increased risk of neurological and cardiovascular disorders and carcinomas [18]. Therefore, high levels of Hcy could be a direct result of vitamin deficiency or enzymatic malfunction [20].

The determination of one-carbon folate metabolites and amino acids in mouse samples is still rare. The content of 5-CH_3_THF found in the mouse plasma in this study was consistent with that reported in the literature as follows: 5-CH_3_THF (0.04 μmol/L) and 5-CHOTHF (0.001 μmol/L) [11]. However, 5-CHOTHF was not detected (N.D.) in our study. The variation between laboratories was mainly due to the different samples and the different analysis methods.

In this study, pABG and hmTHF in mouse plasma had a certain positive correlation, possibly because pABG and hmTHF both reflected the instability of folate. In addition, the total concentration of pABG and hmTHF had a certain positive correlation with that of SAH. To some extent, SAH can reflect the level of Hcy, as the accumulation of Hcy can result in the accumulation of SAH. Abnormally high levels of Hcy in plasma have been identified as an independent risk factor for cardiovascular disease and are also associated with other diseases [36]. In fact, some previous studies have observed that an increased risk of atherogenesis, cancer, or related disease may result from an increased level of hmTHF [37]. Additionally, several studies have suggested that increased folate catabolism can occur in individuals taking oral contraceptives or anticonvulsant drugs, which can increase the risks of related diseases [38]. Because these indicators are all associated with related diseases, a marginally significant higher concentration of SAH was found in plasma samples with higher concentrations of pABG and hmTHF. 

This study comprehensively validated folate extraction from different biological samples including mouse/human WB, plasma, and serum and mouse embryos (this is the first study to determine the folate species in embryos using UHPLC–MS/MS). The UHPLC–MS/MS method developed in this study detected more folate metabolites (eleven compounds) than those currently reported in the literature, as shown in Appendix A [39,40,41,42,43,44]. The LOD and LLOQ (the LOD ranged from 0.17 to 0.68 nmol/L, and the LOQ ranged from 0.55 to 2.25 nmol/L) of the method established in this study were within reasonable ranges and showed that the method was more sensible than the methods proposed by other researchers, as shown in Appendix A [31,39,40,45,46]. The UHPLC–MS/MS method not only can be used for determination in a wide range of matrices (serum, plasma, whole blood, and embryos), but also has good accuracy and precision (intra-day: 92.17–112.53%; inter-day: 92.10–115.29%), and the analysis is quick.

## 4. Materials and Methods

### 4.1. Chemicals

All folate standards employed in this study are shown in Appendix A. LC–MS-grade water, acetonitrile, formic acid, and methanol were obtained from Thermo Fisher Scientific (Fair Lawn, NJ, USA). P5147 protease derived from *Streptomyces griseus* (Type XIV, ≥3.5 units/mg solid, powder, Merck KGaA, Darmstadt, Germany), 10,065 α-Amylase derived from *Aspergillus oryzae* (powder, ~30 U/mg, Merck KGaA, Darmstadt, Germany), ammonium acetate, 2-MCE, and AA were procured from Sigma-Aldrich (St. Louis, MO, USA). Rat serum was purchased from Guangzhou Dongke Biotechnology (production No. BB-71012, Guangzhou, China). Human plasma, mouse serum, mouse plasma, mouse WB, and mouse embryos were obtained from the Medicine Foundation of Peking University. Human serum, 5-CH_3_THF-^13^C_5_, FA-^13^C_5_, and L-Met-^13^C_5_N_5_ were purchased from Sigma-Aldrich (Sigma-Aldrich, Shanghai, China). Human WB was purchased from Shanghai Haopeng Biotechnology (production MA17676, Shanghai, China). 

### 4.2. Standard Solutions

For all standards, the stock solutions for each analyte were prepared at concentrations of 2 mg/mL. For Hcy, SAM, SAH, Met, and L-Met-^13^C5 (isotopic internal standard), they were dissolved in 50:50 (*v*/*v*) methanol/water. For 5,10-CH^+^THF, it was prepared in HCl solution (pH 3) containing 1% AA and 0.2% of 2-MCE. For FA, THF, 5-CHOTHF, 5-CH_3_THF, 10-CHOFA, pABG, 5-CH_3_THF-^13^C_5_ (isotopic internal standard), and FA-^13^C_5_ (isotopic internal standard), they were prepared in NaOH solution (pH 9) containing 1% AA and 0.2% 2-MCE. The calibration curves were constructed as follows: for Hcy, SAM, SAH, and Met, the stock solution was diluted with 50:50 (*v*/*v*) methanol/water with a constant concentration of L-Met-^13^C_5_N_5_ (20 ng/mL); for 5,10-CH^+^THF, it was diluted with 1% AA and 0.2% 2-MCE; for FA, 5-CHOTHF, 5-CH_3_THF, 10-CHOFA, pABG, and THF, it was diluted with 100 mM ammonium acetate buffer (pH 7.85) containing 1% AA and 0.2% 2-MCE with a constant concentration of 5-CH_3_THF-^13^C_5_ (20 ng/mL) and FA-^13^C_5_ (20 ng/mL). The 5-CH_3_THF-^13^C_5_ was used as the internal standard for 5-CH_3_THF and THF. FA-^13^C_5_ was used as the internal standard for FA, 10-CHO-FA, pABG, and 5-CHO-THF. Because there was no commercially available hmTHF standard, the structurally similar 5-CH_3_THF standard was used to quantify it. All the stock solutions and calibration solutions were stored at −80 °C and protected from light. The calibration solutions were freshly prepared because certain metabolites were unstable. 

### 4.3. Sample Preparation

Plasma and serum sample. For plasma preparation, WB was collected from the anticoagulation tube, and it was gently inverted 10 times. It was then centrifuged at 3000 rpm for 10 min at room temperature, and the collected supernatant was further centrifuged at 5000 rpm for 2 min. The supernatant was taken and stored at −80 °C. For serum preparation, after collection of the WB, the blood was allowed to clot by leaving it undisturbed at room temperature for 30 min. Then, it was centrifuged at 3000 rpm for 10 min at room temperature, and the collected supernatant was further centrifuged at 5000 rpm for 2 min before being taken and stored at −80 °C. 

The extraction of folate from serum and plasma was carried out as described by Wang, with some modifications [19]. Briefly, the sample was thawed at room temperature, then 40 μL of 5-CH_3_THF-^13^C_5_, FA-^13^C_5_, and L-Met-^13^C_5_ internal standards, 40 μL of methanol/water (50:50, *v*/*v*) were added to 200 μL of aliquots of plasma or serum and 800 μL methanol containing 50 μg/mL AA and 0.005% 2-MCE as stabilizer. The mixture was vigorously vortexed for 1 min and then centrifuged at 10,000 rpm for 10 min at 4 °C. The supernatant was transferred to a 1.5 mL centrifuge tube and dried down under a gentle stream of nitrogen for about 20 min. The residue was dissolved in 300 μL 100 mM pH 7.85 ammonium acetate buffer containing 1% AA and 0.2% 2-MCE, sonicated for 1 min, and stored at −80 °C for further UHPLC–MS/MS analysis. 

Whole blood sample. The one-carbon folates were extracted from the WB according to the method proposed by Stamm et al. [30]. After thawing the WB for 30 min, 1.2 mL of the sample was sonicated for 3 min. To analyze folate species and total folate, the sample was first treated with 12 μL γ-glutamyl hydrolase (GGH) (0.5 mg/mL) at 37 °C for 30 min to deconjugate the polyglutamyl folate to monoglutamyl folate. To investigate the influence of protein precipitation on the folate recovery, two methods were tested, including methanol precipitation and proteins’ isoelectric point precipitation. For methanol precipitation, 4 mL methanol was added after GGH deconjugation. For proteins’ isoelectric point precipitation, 4 mL 100 mM pH 4.5 ammonium acetate buffer containing 1% AA and 0.2% MCE was added and mixed fully, and the pH of the mixture was adjusted to 4.5. The samples for the two treatments were stirred for 2 min and centrifuged at 10,000 rpm for 10 min at 4 °C. The supernatant was transferred to a centrifuge tube. The methanol precipitation samples were dried under a gentle stream of nitrogen for about 40 min at 40 °C to completely remove the methanol, and then the volume was brought up to 1.5 mL with 100 mM pH 7.85 ammonium acetate buffer containing 1% AA and 0.2% 2-MCE. A 300 μL aliquot of sample was added to 30 μL 5-CH_3_THF-^13^C_5_, FA-^13^C_5_, and 15 μL L-Met-^13^C_5_. All the samples were filtered and stored at −80 °C before UHPLC–MS/MS analysis.

Embryo sample. The method used to extract folates from embryo samples was similar to the method discussed by Belz et al., with some modifications [11]. Embryo samples of approximately 30 mg were frozen at −80 °C and then lyophilized. These were then dissolved by adding 100 μL 100 mM pH 7.85 ammonium acetate buffer containing 1% AA and 0.2% 2-MCE. Then, the samples were sonicated for 2 min to disrupt the cells. To investigate whether endogenous folate was entrapped in the matrix of the embryos, 20 μL di-enzymes of α-amylase (15 U/mL) and protease (50 U/mL) were added to 100 μL embryo extracts and digested at 37 °C for 4 h. Then, 20 μL folate-stripped rat serum was added to the extracts as a conjugase: the mixtures were fully vortexed and incubated for 30 min at 37 °C for deconjugation. To investigate the completeness of deconjugation during incubation, this mixture without protease–amylase treatment was taken out at 30, 60, and 90 min to determine the folate content. After incubation, all samples were heated to boiling point in a water bath to inactivate the enzyme, cooled on ice, and then centrifuged at 10,000 rpm for 10 min at 4 °C.

### 4.4. UHPLC–MS/MS Quantification

Quantitation of folate metabolites was carried out in accordance with the methods described in earlier reports [13]. UHPLC separation was performed using an Agilent 1290 Infinity UHPLC (Agilent, Santa Clara, CA, USA) equipped with a binary pump, autosampler, column oven, and degasser with a Thermo Accucore AQ column (100 mm × 2.1 mm) at 40 °C. The mobile phase was 0.1% formic acid (A) and acetonitrile (B). Its flow rate was 0.3 mL/min, and the sample injection volume was 2 μL. The gradient was as follows: 0–2 min, 5–20% B; 2–4 min, 20–95% B; 4–5 min, 95% B; 5–5.5 min, 95–5% B; 5.5–7.5 min, 5% B. The autosampler was maintained at 4 °C. The UHPLC eluate was introduced into an Agilent 6460 Triple Quad Mass Spectrometer (Agilent, Santa Clara, CA, USA). The MS/MS instrument was operated in the electrospray positive mode. The settings were as follows: capillary voltage: +3.5 kV; nozzle voltage: +1 kV; and cell accelerator voltage: +4 V. Nitrogen was used as the nebulizing gas at 0.31 MPa together with a drying gas flowing at 10 L/min and temperature of 320 °C and a sheath gas flowing at 11 L/min at 380 °C. The nebulizer was set at 45 psi, and the dwell time was 25 ms. An Agilent MassHunter workstation was used to control the equipment as well as for data acquisition and analysis. The quadrupole analyzer was operated at unit mass resolution (0.7 Da), wide resolution (1.2 Da), and widest mass resolution (2.5 Da). Acquisitions were performed via selected reaction monitoring, in which the respective pseudomolecular cation of each folate species was fragmented through collision-induced dissociation (Table 3). Because the mobile phase in this study included 0.1% formic acid, the content of 5,10-CH^+^THF was reflective of the sum of 5,10-CH^+^THF and 10-CHOTHF. 

### 4.5. Method Validation

The developed methodology was validated based on the EMA guidelines, including selectivity, carryover, limits of detection, limits of quantitation, linearity, precision, accuracy, recovery, and matrix effect, as outlined below [19,22]:

Selectivity. The method’s selectivity was examined using analyte-free biological matrices (serum, WB, and plasma); the stripped matrix method involved stirring the extract on ice for 1 h with 100 mg/mL activated charcoal to remove the endogenous folates [13]. Cross-contamination was assessed through the injection of a standard solution of folate and amino acids (at 100 ng/mL) followed by the injection of blank samples. The measured peak area for potential interfering substances had to be less than 20% of the measured peak area for analytes at the LLOQ.

Linearity. Linearity was evaluated from individual standard curves prepared at eight concentrations. The model was considered acceptable if the differences between back-calculated and nominal concentrations were within 15% (20% at LLOQ level).

Limits of detection and quantification. The LLOQ and LOD were first determined by diluting the folate standard solution until the signal-to-noise (*S*/*N*) ratio reached 10 and 3, respectively. The LLOQ was further verified as the lowest concentration of an analyte for which it could be reliably determined that the bias (%) in accuracy and precision were below 20%. The LOD was calculated as one third of the LLOQ.

Precision and accuracy. The intra-day and inter-day accuracy and precision were analyzed using 3 QC samples with 6 replicates. The 3 QC samples with low, medium, and high concentrations are presented in Table 5. Accuracy was considered acceptable when the determined concentration was within ±15% of the nominal concentration for all QC samples. These reference materials were measured in triplicate in three runs. Similarly, the precision was considered suitable when the coefficient of variation of the replicates was lower than ±15% for the QC samples. The recovery was calculated using the following equation:(1)$$Recovery\;\left(\%\right)=\frac{\left(A-B\right)*100}{C}$$
where A is the analyte concentration in sample extracts spiked with individual QC standard solutions, B is the analyte concentration in the sample extracts, and C is the nominal value of different analytes in the standard solution. The analyte concentration obtained from A and B was calculated against the calibration curve.

Matrix effect. The matrix effect was also determined for the method. The sample matrix, such as co-eluting compounds, can contribute to alterations in the analyte ionization and overall response. Therefore, complete separation of the analyte and the co-eluting matrix could help to decrease or increase the ionization of the target analyte. The matrix effect was determined using the following method: 10 μL QC at a concentration of 3 LLOQ was spiked into 1 mL post-sample extract. The peak area obtained in the matrix was compared with the corresponding peak area in the pure solvent with the same concentration. Each sample was analyzed three times. The matrix effect was calculated as follows:(2)$$Matrix\;effect\;(\%)=\frac{Y}{X}*100\%$$
where X is the endogenous analyte peak area in the pure solvent, and Y is the analyte peak area in the sample matrix with the same amount of analyte added.

### 4.6. Calculation of Red Blood Cell (RBC) Folate Concentrations

RBC folate concentrations were calculated according to the method proposed by Lamers et al. [47]:(3)$$RBC\;folate=\frac{\left(whole\;blood\;folate\times 100)-[plasma\;folate\times (100-hematocrit)\right]}{hematocrit}$$

### 4.7. Data Processing and Statistical Analysis

One-way ANOVA analysis of variance between groups was performed by IBM SPSS Statistics 26.0. ChemDraw22.0, Origin Pro 2021, and GraphPad Prism 8.0 were used for plotting the figures. Data results are expressed as mean ± standard deviation.

## 5. Conclusions

A method involving simple and rapid sample preparation and a short UHPLC–MS/MS runtime was developed and validated for the simultaneous determination of one-carbon-related folate metabolites and amino acids in various biological samples. The short time required for sample preparation (about 10 min), with the exception of the embryo incubation (30 min), and UHPLC–MS/MS (7.5 min/sample) will facilitate folate measurements in large-scale studies during routine analysis. Methods for the extraction of folate from embryos and WB were optimized. After optimization, a total deconjugation time of 30 min and acidification to a pH below 4.5 were the most suitable methods for the extraction of one-carbon-related folate metabolites and amino acids from embryos and WB, respectively.

## Figures and Tables

**Figure 1 ijms-25-03458-f001:**
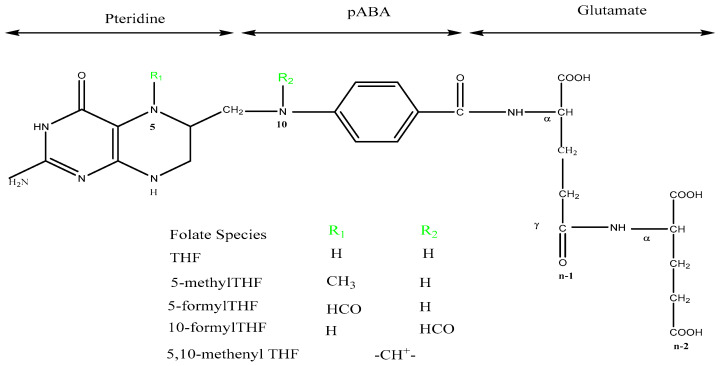
Chemical structures of folate species.

**Figure 2 ijms-25-03458-f002:**
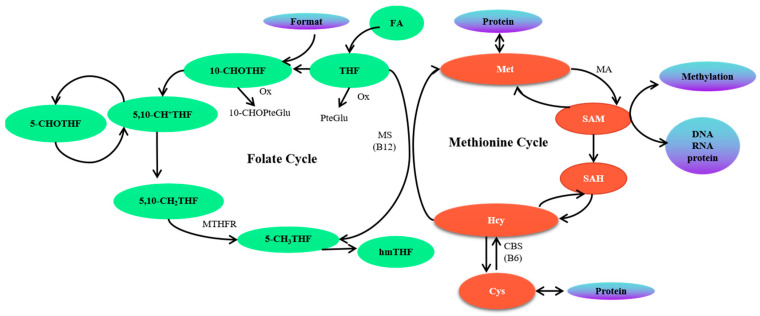
Summary diagram of folate one-carbon metabolism. MS: methionine synthase; CBS: cystathionine β-synthase; MTHFR: methylene-tetrahydrofolate reductase; Ox: oxidation; MA: methionine adenosyltransferase.

**Figure 3 ijms-25-03458-f003:**
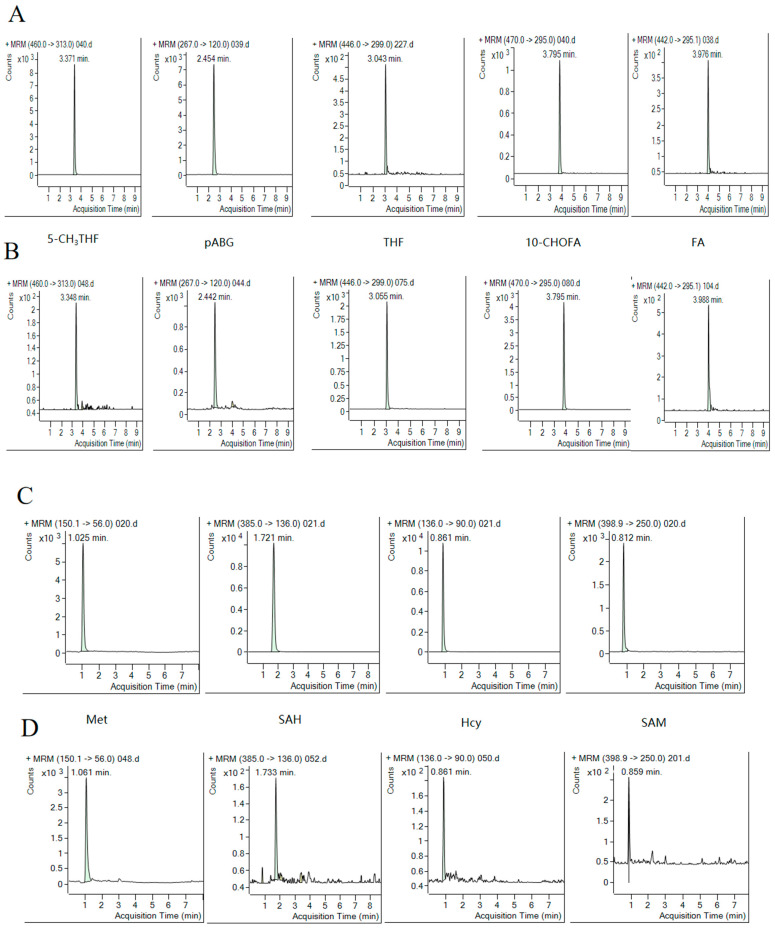
A typical UHPLC–MS/MS chromatogram showing one-carbon-related folate metabolites in human plasma: (**A**) Folate metabolite standards. (**B**) Folate metabolite in human plasma samples. (**C**) Amino acids related to folate metabolism standards. (**D**) Amino acids related to folate metabolism in humans.

**Figure 4 ijms-25-03458-f004:**
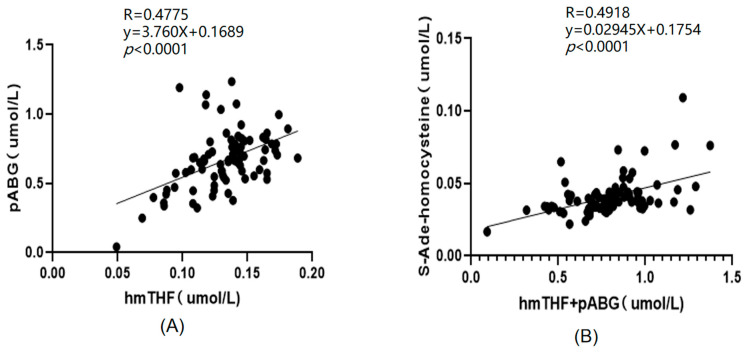
Correlation analysis of folate metabolite concentrations in rat plasma. (**A**) Analysis of correlation between hmTHF and pABG concentrations. (**B**) Analysis of correlation between hmTHF+pABG and SAH concentrations.

**Table 1 ijms-25-03458-t001:** The distribution of folate and amino acids in embryos (ng/g).

	S-30 ^1^	S-60	S-90	Protease–Amylase ^2^
5,10-CH^+^THF	103.8 ± 12.2 a	112.1 ± 0.6 a	120.4 ± 9.4 a	N.D.
pABG	15.0 ± 5.2 a	12.7 ± 1.2 a	12.7 ± 2.9 a	33.7 ± 3.5 b
5-CH_3_THF	45.5 ± 7.9 a	24.0 ± 0.8 b	12.5 ± 0.4 c	N.D.
Met	7863.3 ± 179.0 a	8194.5 ± 118.1 a	8162.2 ± 266.2 a	(31.3 ± 2.9) × 10^3^ b
SAH	1629.2 ± 38.5 a	1094.1 ± 19.8 b	742.1 ± 91.1 c	20.4 ± 1.0 d
SAM	246.2 ± 26.7 a	217.3 ± 10.7 ab	204.1 ± 18.4 a	N.D. ^2^

^1^ S-30, S-60, and S-90: embryos were subject to a deconjugation time of 30, 60, and 90 min. ^2^ Protease–amylase: di-enzymes of α-amylase (15 U/mL) and protease (50 U/mL). Values are given as the mean ± 2 standard deviations (*n* = 3) [25]. Values marked with lowercase letters (a–d) indicate that the incubation times (30–90 min) and double enzymatic hydrolysis were significantly different for each compound (*p* < 0.05). N.D.: Not detected.

**Table 2 ijms-25-03458-t002:** The distribution of folate and amino acids in whole blood (nmol/L).

	Experiment 1 ^1^	Experiment 2 ^2^
pABG	46.5 ± 2.7 b	91.4 ± 4.8 a
5-CH_3_THF	25.3 ± 3.1 b	48.2 ± 1.9 a
10-CHOFA	10.1 ± 1.4 b	39.8 ± 2.9 a
FA	74.9 ± 11.5 b	132.2 ± 16.8 a
hmTHF	10.9 ± 2.0 a	N.D.
Hcy	12.8 ± 1.4 a	11.3 ± 2.7 a
SAH	59.7 ± 9.1 b	80.5 ± 13.8 a
Met	1199.3 ± 38.3 b	2476.4 ± 164.0 a

^1^ Experiment 1: methanol precipitation. ^2^ Experiment 2: proteins’ isoelectric point precipitation. Values are given as the mean ± 2 standard deviations (*n* = 3) [27], and values marked with different lowercase letters (a,b) indicate that folate and amino acids in WB after extraction in Experiments 1 and 2 were significantly different (*p* < 0.05). N.D.: Not detected.

**Table 3 ijms-25-03458-t003:** Instrument settings ^1^.

Internal Standard	Type	Transition (*m*/*z*)	CE (V)	Fragmentor (V)
5-CHOTHF	Quantifier	474→327	19	140
	Qualifier	474→299	45	140
pABG	Quantifier	267→120	15	90
	Qualifier	267→130	15	90
5-CH_3_THF	Quantifier	460→313	19	140
	Qualifier	460→400	8	140
10-CHOFA	Quantifier	470→295	24	160
	Qualifier	470→323	15	160
FA	Quantifier	442→295	16	110
	Qualifier	442→176	53	110
510-CH^+^THF	Quantifier	456→412	33	180
	Qualifier	456→327	33	180
THF	Quantifier	446→299	16	110
	Qualifier	446→166	59	110
Met	Quantifier	150→104	5	80
	Qualifier	150→56	15	80
SAH	Quantifier	385→136	25	70
	Qualifier	385→134	35	70
SAM	Quantifier	399→250	15	120
	Qualifier	399→298	27	120
Hcy	Quantifier	136→90	5	80
	Qualifier	136→118	20	80
5-CH_3_THF-^13^C_5_	Quantifier	465→313	19	140
FA-^13^C_5_	Quantifier	447→295	16	110
L-Met-^13^C_5_	Quantifier	155→104	5	80

^1^ Abbreviations: CE, collision energy; 5-CH_3_THF-^13^C_5_, 5-methyltetrahydrofolic acid-^13^C_5_; FA-^13^C_5_, folic acid-^13^C_5_; L-Met-^13^C_5_, L-methionine-^13^C_5_.

**Table 4 ijms-25-03458-t004:** Linearity ranges, limits of detection, and quantitation of the eleven compounds.

Compound	Linearity Range (nmol/L)	LOD (nmol/L)	LOQ (nmol/L)
5-CHOTHF	0.47–47.4	0.32	1.06
pABG	0.267–26.7	0.28	0.94
5-CH_3_THF	0.69–46.0	0.33	1.09
10-CHOFA	0.47–47.0	0.32	1.07
FA	0.44–44.2	0.31	1.02
510-CH^+^THF	0.46–45.6	0.17	0.55
THF	0.89–44.6	0.68	2.25
Met	0.075–15.0	0.33	1.10
SAH	0.19–38.5	0.20	0.65
SAM	0.60–39.9	0.38	1.26
Hcy	0.14–13.6	0.56	1.85

**Table 5 ijms-25-03458-t005:** Precision and accuracy of the UHPLC–MS/MS method.

Compounds	Concentration (μmol/L)	Precision (%)		Accuracy (%)	
		Intra-Day	Inter-Day	Intra-Day	Inter-Day
5-CHOTHF	Low (0.02)	3.76	7.11	107.88	100.82
	Medium (0.04)	3.05	8.40	98.68	104.73
	High (0.10)	2.34	9.87	99.56	98.19
pABG	Low (0.04)	5.55	6.32	103.86	110.59
	Medium (0.07)	6.80	8.43	101.29	107.52
	High (0.19)	4.31	9.19	112.53	97.17
5-CH_3_THF	Low (0.02)	3.05	10.5	104.40	104.42
	Medium (0.04)	5.57	6.70	103.19	108.99
	High (0.10)	3.41	4.83	107.73	103.62
10-CHOFA	Low (0.02)	4.33	12.04	104.34	92.85
	Medium (0.04)	6.35	8.35	97.03	106.63
	High (0.11)	4.85	4.98	105.58	106.81
FA	Low (0.02)	5.32	9.08	106.23	114.45
	Medium (0.05)	2.51	8.53	103.18	97.13
	High (0.11)	6.48	10.9	96.40	99.64
510-CH^+^THF	Low (0.02)	1.37	5.07	96.08	108.23
	Medium (0.04)	3.35	10.9	100.13	107.57
	High (0.11)	2.15	5.21	106.78	97.97
THF	Low (0.07)	4.58	7.07	105.20	92.82
	Medium (0.13)	3.74	9.42	98.84	92.36
	High (0.20)	5.34	6.81	100.09	102.34
Met	Low (0.2)	7.55	9.62	95.88	115.29
	Medium (0.4)	5.20	9.19	93.83	97.88
	High (0.6)	1.01	0.69	97.05	99.91
SAH	Low (0.05)	1.43	7.33	95.64	105.87
	Medium (0.10)	5.11	9.08	92.17	93.91
	High (0.21)	6.57	9.05	97.46	100.94
SAM	Low (0.05)	3.37	1.07	100.21	101.25
	Medium (0.10)	4.60	3.88	96.32	109.48
	High (0.20)	1.09	1.98	102.48	92.10
Hcy	Low (0.15)	1.53	10.8	102.58	96.36
	Medium (0.29)	3.64	4.92	98.99	104.45
	High (0.59)	3.36	4.35	97.55	97.03

**Table 6 ijms-25-03458-t006:** Distribution of folate and amino acids in different biological samples ^1^.

	Mouse WB (*n* = 3)	Mouse Serum (*n* = 3)	Mouse Plasma (*n* = 80)	Mouse Embryos (*n* = 24)	Human WB (*n* = 3)	Human Serum (*n* = 3)	Human Plasma (*n* = 480)
5,10-CH^+^THF	123.4 ± 21.3	N.D.	N.D.	74.3 ± 14.4	N.D.	N.D.	N.D.
pABG	1051.1 ± 41.5	171.3 ± 45.3	864.0 ± 166.1	20.6 ± 10.4	92.7 ± 7.9	74.0 ± 10.6	1192.0 ± 202.4
5-CH_3_THF	338.3 ± 68.7	70.1 ± 1.7	202.2 ± 165.5	185.4 ± 36.7	54.1 ± 10.2	38.8 ± 1.6	13.6 ± 2.4
hmTHF	200.1 ± 62.2	60.3 ± 10.3	122.2 ± 101.0	N.D.	N.D.	12.3 ± 1.1	24.6 ± 1.2
10-CHOFA	N.D.	N.D.	N.D.	N.D.	44.1 ± 2.0	N.D.	50.4 ± 16.2
FA	35.3 ± 20.3	N.D.	N.D.	N.D.	131.1 ± 20.8	N.D.	23.8 ± 10.4
Met	(196.2 ± 21.2) × 10^3^	1880.3 ± 337.4	(8.6 ± 1.4) × 10^3^	(16.5 ± 3.2) × 10^3^	2482.3 ± 83.7	3429.4 ± 226.6	(20.5 ± 17.0) × 10^3^
SAH	1550.7 ± 133.2	31.4 ± 16.4	60.6 ± 10.4	1093.2 ± 216.1	81.4 ± 6.8	68.0 ± 10.4	39.2 ± 3.2
Hcy	N.D.	146.2 ± 18.3	N.D.	N.D.	12.8 ± 2.9	234.4 ± 23.4	643.3 ± 104.1
SAM	835.2 ± 9.2	N.D.	N.D.	132.8 ± 26.6	N.D.	N.D.	381.7 ± 52.4

^1^ The unit for mouse embryos is ng/g, and for the others, it is nmol/L. N.D.: Not detected.

## Data Availability

The data presented in this study are available on request from the corresponding author.

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
