# Peer review of "Simultaneous Determination of One-Carbon Folate Metabolites and One-Carbon-Related Amino Acids in Biological Samples Using a UHPLC–MS/MS Method"

_ijms, 2024, doi:10.3390/ijms25063458_

Round 1

Reviewer 1 Report (Previous Reviewer 2)

Comments and Suggestions for Authors

Although I had already called for a significant improvement in the fluidity and clarity of the text, avoiding repetition and redundant sentences, there are still several problems. Moreover several mistakes and unclear definitions are found in the new version of manuscript.

For example:

-Lines 65-76

 “Some biological samples such as serum, plasma and WB are often used as indicators for monitoring folate status in test subjects. Folate levels in different biological samples give different information at clinical level. Folate in serum is considered as an indicator of recent folate intake because it has no established cut-off serum folate for neural tube defect risk [7]. WB can reflect the long-term folate status with a biological half-life of 8 weeks and is generally used to predict the risk of neural tube defects [8]. Folates in plasma are well established biomarkers of diseases and can reflect dietary folate intake [9]. Additionally, in early studies involving pregnant mice with neural tube defects, an alternation of the folate pattern in the embryo has been observed and therefore of folate variations in embryos also merit investigation [10].”

This part should be better written since there are repetitions and some confusing sentences. For example, this part could be changed as follows:

“Folate levels in different biological samples such as serum, plasma and WB provide different information at the clinical level. Serum folate is considered an indicator of recent folate intake, as there is no established serum cut-off for the risk of neural tube defects [7]. On the other hand, folate concentrations in WB may reflect long-term folate status with a biological half-life of 8 weeks and then this parameter is generally used to predict the risk of neural tube defects [8]. Finally, plasma folate levels are well-known biomarkers of disease and may reflect dietary folate intake [9]. Furthermore, in early studies involving pregnant mice with neural tube defects, an alteration in the folate pattern of the embryo was observed and therefore also this aspect merits investigation [10].

- lines 68-69. I do not understand the mean of the following sentence:

“Folate in serum is considered as an indicator of recent folate intake because it has no established cut-off serum folate for neural tube defect risk [7]”

What is the connection between the lack of a cut-off and that serum folate levels indicate a recent intake?

Other imprecisions and grammar errors are present in the manuscript. Please rewrite all the text.

-Lines 221-222

“Selectivity met the criterion because there were no interfering peaks observed at the same retention times as the target analytes.”

Should be rewritten (for example):

Selectivity met the criterion because no interfering peaks were observed at the same retention times of target analytes.”

Lines 221-227

“By following the determination of folate and amino acids, there was no interference in the stripped matrix and no obvious cross-contamination existed after injection with 100 ng/mL standard solution”

Should be rewritten (for example):

“By following the determination of folates and amino acids, there was no interference in the stripped matrix and no obvious cross-contamination existed as demonstrated injecting a 100 ng/mL standard solution”

-Lines 249-250

The sentence:

“For trueness, the intra-day and inter-day variation are 92.17 to 112.53%, 92.10 to 115.29%, respectively”.

Should be rewritten:

“For trueness, the intra-day and inter-day ranges were from 92.2 to 112% and from 92.1 to 115 %, respectively”

Those reported by the authors can be better described such as “range” (interval).  Note, the number of significant figures is too high. Please reduce to maximum three. In my opinion considering five significant figures (eg: 112.53%) are not acceptable.

Lines 545-552

“The recovery was calculated using the following equation:

Recovery (%)=(A−B)100/C                                           ( I )

where A is the analytes concentration in sample extracts spiked with individual QC standard solutions, B is the folate analytes concentration in the sample extracts; and C is the nominal value of different analytes in the standard solution. The analytes concentration obtained from A and B was calculated against the calibration curve”.

Excuse me, but I am still unclear about the difference between “A” and “B”.

- Lines 561-567

The equation (II) for the calculation of matrix effects is not readable. Moreover in this second equation “A” and “B” must correspond to those previously introduced (see equation (I)); otherwise different capital letters must be used.

-Tables 1 and 2

Several data in the Tables (mainly in Table 2) are not correctly reported, because the number of digits after the decimal point of the mean and the standard deviation must be coherent. Authors should be familiar with these basic rules of analytical chemistry. See for example the Eurachem/CITAC Guide “Guide to Quality in Analytical Chemistry. An Aid to Accreditation”. Third edition (2016) (available at: www.eurachem.org):

“16.13 The significant figures used to report the measurement result and its uncertainty should be consistent with the measurement capability. Therefore, in most analytical measurements, values for the expanded uncertainty should be reported with no more than two significant digits. The measurement result should be rounded [55] to be consistent with the stated uncertainty. For example, given a result of 215.342 mg kg-1 with an estimated combined standard uncertainty of 5.12 mg kg-1, which corresponds to an expanded uncertainty of 10.24 mg kg-1, the reported result should be: 215 mg kg-1 ± 10 mg kg-1 (k = 2, 95% confidence level).”

Therefore, a result furnished as (0.05±0.004) is wrong. Correct in the whole manuscript.

I also suggest reading the document: “REPORT ON THE RELATIONSHIP BETWEEN ANALYTICAL RESULTS, MEASUREMENT UNCERTAINTY, RECOVERY FACTORS AND THE PROVISIONS OF EU FOOD AND FEED LEGISLATION, WITH PARTICULAR REFERENCE TO COMMUNITY LEGISLATION CONCERNING” (https://food.ec.europa.eu/system/files/2016-10/cs_contaminants_sampling_analysis-report_2004_en.pdf)

-Table 4

The linearity range and LOD/LOQ should be furnished in the same unit (nmol/L)

-Table 6

Data furnished as follows: “0.09±0.00” are unacceptable since all the measurement are affected by a measurement error (standard deviation, uncertainty …). Therefore, the standard deviation cannot be equal to zero!

-Figure 3

I requested several times to authors the revision of Figure 3 splitting it in two new Figures. The chromatograms are too small and the reader cannot evaluate them. Furthermore, the vertical red line completely covers the peaks. Please revise it.

Comments on the Quality of English Language

The manuscript needs a significant improvement of the fluidity and clarity of the text, avoiding repetitions and redundant sentences which are still present. Moreover also some grammar mistake has been found.

Author Response

Reviewer 2 Report (Previous Reviewer 1)

Comments and Suggestions for Authors

The manuscript describes a novel MS-based methodology of determination on molecular species related to folate metabolism and selected amino acids, which were validated and applied for analysis of various biological samples. The manuscript has already been corrected after a series of my revision reports. Currently, I have only some minor remarks.

Line 222 – The method of obtaining of analyte-free biological matrices should be described in the Materials and Methods section instead of the Results.

Supplementary Table 3/2 “Detection and quantitation limits of folate metabolites in the literature” – please correct the mistake in the unit ‘nmlo/L” -> ‘nmol/L”

Please double-check the table numbering in the Supplementary materials

Author Response

Reviewer 3 Report (New Reviewer)

Comments and Suggestions for Authors

Author Response

Reviewer 4 Report (New Reviewer)

Comments and Suggestions for Authors

This article is covering specific aspects of possible development of a simple fast and sensitive ultra-high-performance liquid chromatography MS/MS (UHPLC MS/MS) method for the simultaneous quantification of folate metabolites.  

Folic acid (FA) tetrahydrofolic acid (THF) and other derivatives along with homocysteine, methionine can be analyzed in the biological samples as well. The measured contents of folate in human plasma, serum and whole blood can be effectively detected in variety of concentrations. The sample preparation via deconjugation of polyglutamylfolate  by incubation with striped rat serum is essential for accurate determination and precise analysis of all biological samples.

 The data collection and quantification of analyzed compounds are compiled in Table1-6. Additionally, compilation of important data on measurements and correlation analysis of folate metabolite concentration in plasma 

is presented in figure 4. Figure 2 is also critically important for understanding folate one-carbon metabolism. This will constitute crucially important goals and novelty of this important paper. 

The following suggested small changes and recommendations should be introduced before the publication of the manuscript.

  • Page 2. Line 56. Insert “Homocysteine” in front of “Hcy” in parenthesis. 
  • Page 3. line 75. Insert “whole blood” in front of “WB” in parenthesis. 
  • Page 12. Line 331. Replace “research “with  “study”. 
  • Page 13. Line 386. Replace “are” with ‘were”.
  • Page 14. Correct the reference Stamm et al to [27].
  • Page 15. Line 492. Correct “coeluting” to “co-eluting”.
  • Page 16. Line 503. Insert the literature reference number for Lamers et al: [x] 

The manuscript is of good quality and urgent importance and is written and edited in order to meet the standard for the articles published in International Journal of Molecular Sciences. Thus, I certainly recommend it for publication after the correction of these suggested minor changes. 

Round 2

Reviewer 1 Report (Previous Reviewer 2)

Comments and Suggestions for Authors

As already noted several times during this review, in my opinion the authors should improve their analytical skills, as they present a work that is largely based on the validation of a quantitative method. I had asked for some points to be revised, but then new errors are added.

For example in Table 6, the authors have corrected the problem of standard deviations equal to zero, but then they wrongly truncate the decimal digits.

For example (5,10-CH+THF  in mouse WB):

123 ± 21.3

should instead be:

123±21

In addition as I have already said many times, reading the revised manuscript is very difficult because the deleted and corrected parts both remain visible and this creates confusion.

Comments on the Quality of English Language

English Language is understandable although it could be made more fluent

Round 3

Reviewer 1 Report (Previous Reviewer 2)

Comments and Suggestions for Authors

That's enough for me. The manuscript must be rejected for the reasons that I have explained on several occasions.

Comments on the Quality of English Language

no comment

Author Response

No comment.

This manuscript is a resubmission of an earlier submission. The following is a list of the peer review reports and author responses from that submission.

Round 1

Reviewer 1 Report

Comments and Suggestions for Authors

The manuscript describes a new MS-based methodology of determination on molecular species related to folate metabolism and selected amino acids, which were validated and applied for analysis of biological samples. The paper provides some interesting pieces of information, however, there are some uncertainties and gaps that need to be filled before publication. My remarks are listed below:

Line 74 – exaction? or extraction?

Line 129 – “Since the extraction method of folate from plasma and serum has been extensively evaluated in other studies, these were not optimized again. “– References should be provided.

Line 152 – “…there were no significant difference for 5, 10-CH+THF, pABG, Met and SAM” – Was this checked by statistical tests?

Table 1 – Please explain what letters “a”, “b”, “c”, “d” mean.

Table 2 – Please explain what letters “a”, “b” mean.

Line 160 – “which has been fully optimized in early studies.” – What studies? Please provide the reference.

Line 168 – “the folate metabolites concentrations in method 1 were significantly different from concentrations in method 2.” – Was this checked by statistical tests? Every time the Authors use “significant” it should be proved by statistics.

Figure 3 - Why was this particular type of biological sample chosen to present the chromatogram of a particular metabolite? Another remark concerns the missing chromatograms of some analytes, e.g. folic acid, please add them to the Figure.

Line 180 - The Authors wrote: “The most abundant fragment ion was chosen as quantifier, and the second most abundant as qualifier. For internal standards, only one mass transition was used in order to increase the sensitivity of the method towards analytes (Table 3).” However, based on data contained in Table 3 the above-cited statements seem untrue. For seven out of 11 analytes only one MRM transition was monitored. Whereas for two out of three internal standards two MRM transitions were listed, not one. Moreover, internal standards should be named “internal standards” instead of “analytes”.

“Method selectivity was assessed by analyzing a folate-stripped blank.” – blank matrix? What type of matrix? Serum, plasma, embryo or WB?

Table 4 – Could the Authors provide linearity ranges for analytes also in umol/L?

Line 186 – “Quantitation was performed based on retention time, which was compared with the authentic standard MRM transition” – I do not agree with this statement. Quantitation was performed based on peak areas of analytes and their corresponding internal standards, and linear regression model, whereas peak identity was confirmed based on agreement with standards’ retention time and MRM transitions.

Table 8 – For mouse plasma, mouse embryos, and human plasma, the Authors indicated in brackets the number of samples analyzed. The number of samples analyzed should be provided also for the remaining biological matrices.

Why Table S1 is incorporated in the main text of the manuscript?

The Authors concluded that: “The short time required for sample preparation (about 10 min)…” whereas a deconjugation time of 30 min was required in the case of WB. Please correct this conclusion.

Comments on the Quality of English Language

English language should be corrected, there are many errors, including grammar ones, e.g.

·      UHPLC-MS/MS methods which are rapid, high sensitivity and selectivity - are rapid, highly sensitive and selective

·      Analytes of folate species and related amino acids were detected in positive ion mode and was well-separated on the reversed phase …- were well-separated

Reviewer 2 Report

Comments and Suggestions for Authors

The work describes an interesting topic, but there are several major concerns:

1-There is not novelty since sample preparation of plasma/serum is not original (see reference 14). The chromatographic approach (reversed phase separation) does not seem suitable for the involved analytes since some of them are not retained.

2 - The validation study and results are confusing and redundant. In some cases the reported results (see e.g. precision data) are not plausible. In addition the use of significant figures is inappropriate (too many figures) demonstrating an insufficient knowledge of basic rules of analytical chemistry

3 - The text is not well organised.

According I cannot recommend the publication on International Journal of Molecular Sciences.

 Some more detailed comments:

Lines 185-186 “All of the molecules exhibited adequately separated chromatographic peak that 185 was easily distinguished from the baseline as shown in Figure 3”. This affirmation it is not true: for example 5-CHOTHF and 510-CH+THF were not separated (see Table 3) as demonstrated by their retention times (3.694 min and 3.656 min, respectively).

Lines 205-206 “All the calibration curves showed good linearity (0.992≤R2≤0.999) and were suitable for analysis of the samples”. R2 is not a linearity index. Moreover in Section 4.5 (lines 452-454) the authors claim “Linearity was evaluated from constructed the standard curve with eight concentrations of individual folate species as detailed earlier. For the selected model to be acceptable, back-calculated mean concentration were to be within 15% of the nominal value (20% at LLOQ level)”. Therefore it is not clear what is the used criterion to assess linearity: R2 or back-calculated mean concentration?

Lines 461-463: The validation study must be better described. “The accuracy and precision were analyzed with 6 replicates of quality control samples spiking with low, medium, and high concentration points in extracts of folate-stripped rat serum and the results of accuracy and precision are presented in Table 5”. Weren't experiments done also on different validation days? On the other hand, in Table 5 intra and inter day precision and accuracy are reported and therefore these parameters seem also tested during different days (inter day experiments).

Table 3: the retention times are furnished with too much significant figures! For SAH, SAM and  Hcy the qualifier ions are not reported

Table 3. The retention times of some analytes are very short (< 1.5 min). Do authors measured the retention time of a compound not retained to ascertain this aspect? I think that the applied LC column (Accucore aQ, Thermo Scientific) based on reversed phase mechanism could be not suitable. This is also confirmed by a previous paper of the same research group who applied a HILIC column to separate folate metabolites (see reference 11: Zou et al. Quantification of polyglutamyl 5-methyltetrahydrofolate, monoglutamyl folate vitamers, and total folates in different berries and berry juice by UHPLC-MS/MS. Food Chem. 2019, 276, 1-8)

Table 4: some information in this table are not useful (slope, R2). Please delete these colums. LOD and LOQ are “instrumental” LOD and LOQ? Please detail.

Table 5. “Precision and accuracy of the LC-MS/MS method”. the use of term “accuracy” is discouraged (VIM - International Vocabulary of Metrology – Basic and General Concepts and Associated Terms. Third Edition, 2007). Please change accuracy with trueness.

Table 5. the unit “pg/injection” is not suitable. Please convert in “μg/g” and “μmol/L” coherently with the units used in Table 8.  By the way, the injection volume is not reported in “Materials and Methods” section.

Table 5. The CVs obtained in inter-day precision conditions are generally lower than CV obtained in inter-day conditions. In Table 5 intra-day CVs are generally higher than inter-day CVs. See also the intervals reported for method precision in lines 217-219. These results are suspect.

Tables 6 and 7 are redundant since this information is not fundamental. They must be summarised in the text. Recoveries in Table 7 are absolute recoveries? Please specify.

Table 8 must be revised. In some cases ranges are furnished, in other cases a value (mean) +/- standard deviation(?). In addition “0” must be replaced with “not detected”.

Figure 3: the reported chromatograms are too much and too little to see. This Figure needs to be completely redone.

Comments on the Quality of English Language

The use of English is often inappropriate and, in some cases, this makes comprehension of the text complicated because the concepts are not stated in appropriate sentences.

Some examples:

Line 391: The sentence “The blood of healthy volunteers is collected using traditional intravenous blood sampling..” should be changed as follows “The blood of healthy volunteers was collected using traditional intravenous blood sampling…”

Another example (lines 452-454). The sentence “Linearity was evaluated from constructed the standard curve with eight concentrations of individual folate species as detailed earlier. For the selected model to be acceptable, back-calculated mean concentration were to be within 15% of the nominal value (20% at LLOQ level)” is not well written. For example it could be rephrased as follows: “Linearity was evaluated from individual standard curves prepared at eight concentrations. The selected model was considered acceptable if the differences between back-calculated and nominal concentration were within 15% (20% at LLOQ level)”.

Round 2

Reviewer 1 Report

Comments and Suggestions for Authors

 The Authors corrected the manuscript according to the remarks. However, I still have some other remarks regarding the revised version of the manuscript.

-             Novelty of the method – After reading the revised version of the manuscript, I have an impression that this work shows little novelty. Discussion on the significance or novelty of this work is needed before it can be considered for publication. Please compare the developed method with other LC-MS/MS methods of determination of folate metabolites. Especially please compare the LOD and LOQ values of the presented method with the LOD and LOQ values of the methods in the literature as “challenge associated with the quantification of endogenous folate in different biological samples is their presence in extremely low concentrations.”

-             Table 4 - Are the values instrumental LOD and LOQ or method LOD and LOQ?

-             Line 87: “For decades, various analytical methods have been developed such as microbiological assay, radioimmunoassay, capillary electrophoresis and UHPLC–MS/MSUHPLC-MS/MS [2].” – Giving only one reference is not enough. Moreover, cited reference no. [2] refers to the study of atherosclerosis (“Epigenetic factors in atherosclerosis: DNA methylation, folic acid metabolism, and intestinal microbiota”) and is not a methodological publication nor review article of the methods.

-             Line 216 – “The method involved quantification of endogenous internal standards, and by definition, internal standards-free biological matrices do not exist for endogenous compounds. “ – In this sentence, the word “analyte” has been incorrectly and unnecessarily replaced by the “internal standard”. This made that sentence meaningless. I haven’t heard about “, internal standards-free biological matrices” before. In the determination of endogenous compounds, a lack of analyte-free biological matrices is a challenge. The internal standard is usually a compound not present in a sample. Do the Authors have their own interpretation of this issue? In my previous review, I wrote: “Moreover, internal standards should be named “internal standards”

instead of “analytes” “ and this remark referred to Table 3 (and it should still be addressed). In the cover letter, the Authors wrote: “The internal standard has been named as “internal standards” instead of “analyte” as shown in page 6 line 199 and page 7 line 208-209.” However, I have a problem finding these changes, probably, the line numbers have changed in the version of the manuscript with tracked changes. Please cite these sentences in the cover letter.

Reviewer 2 Report

Comments and Suggestions for Authors

The revision of manuscript has improved some aspects, but its quality remains largely insufficient.

The authors' responses to my comments were in some cases deficient (see, for example, observation Q5 and Q9) or misrepresented. For example, with my remark "The validation study and results are confusing and redundant. In some cases the reported results (see e.g. precision data) are not plausible. In addition the use of significant figures is inappropriate (too many figures) demonstrating an insufficient knowledge of basic rules of analytical chemistry" I did not mean the "figures" as pictures, but the significant figures (digits).

Round 3

Reviewer 1 Report

Comments and Suggestions for Authors

I am satisfied with the Authors' response and the revised version of the manuscript.

The list of novelty aspects of the presented method (response 1) should be provided in the manuscript in the discussion section.

Reviewer 2 Report

Comments and Suggestions for Authors

The revision of manuscript is insufficient to improve it significantly. Some grammar errors have been corrected or other minor problems, but the text is sometimes unintelligible, inadequate, and, often, the authors use non-specialist language. The descriptions of experimental steps are approximate.  As I already stated, the manuscript has substantial problems that cannot be corrected by revising a few sentences as suggested by the reviewers. These problems are due, in my opinion, to the scarce knowledge of analytical chemistry concepts and terminology. Some examples:

-Lines 65-66- “Different biological samples have different clinical purposes”.

I understand what the authors mean, but it is expressed very poorly. The folate concentrations give different information, not “biological samples”!

I suggest for example “Folate levels in different biological samples give different information at clinical level”

-Lines 75-76. “Different biological samples are likely to require different methodologies for extraction of the endogenous folate”. This sentence expresses a completely obvious concept.

-Lines 153-155. “In terms of hydrolysis by protease–amylase protease-amylase, this treatment caused a significant loss of 5,10-CH+THF and 5-CH3THF to below the level of detection (LOD).”

This sentence should also be written better. For example:

“Considering the hydrolysis by protease-amylase, this treatment caused a significant loss of 5,10-CH+THF and 5-CH3THF whose concentrations were no longer detectable (<LOD)”

-Line 192. “2.2. Method validation” instead of “2.2. MS/MS Validation”. The whole procedure is validated, not only the instrumental determination

-Lines 233-234. “The LOQ were set as the lowest calibration samples showing acceptable trueness (100±20%), precision (RSD≤20%), and signal-to-noise ratio (≥10).” This sentence should be rewritten.

For example: “LOQs were set as the lowest calibration concentration showing acceptable trueness (100±20%), precision (RSD≤20%), and signal-to-noise ratio (≥10).” LOQ were set at the concentration of the lowest calibration sample, they were not set as “the lowest calibration samples”.

Lines 222-224. “By following the determination of folate and amino acids, there was no interference in the stripped matrix and no obvious cross-over existed after injection with 100 ng/mL standard solution.”

By "cross-over," do the authors mean "cross-contamination"?

Lines 481-482. “The Q Quadrupole analyzer operated at unit mass resolution (0.7 amu), and also at wide resolution (1.2 amu) and the widest mass resolution (2.5 amu).”

Resolution at 1.2 amu and 2.5 amu are lower resolution than at 0.7 amu. In addition, “amu” is an obsolete unit in mass spectrometry.

-Lines 492-493. Selectivity. “Selectivity was determined by analyzing the three different folate-stripped samples (serum, WB and plasma) and subjecting them to the same sample preparation”.

What does it means? This sentence is incomplete and I cannot understand how authors evaluate selectivity in practice. They compare the different matrix types and evaluate the potential interfering substances? How? Probably “samples” in this context means “matrices” (serum, WB, plasma)

Line 494-495 “Cross-over was evaluated by injection with the folate and amino acids standard solution (100 ng/mL) and then injection with the blank samples.”

This sentence is not well written. It can be re-written, for example, as follows:

-Line “Cross-over was assessed by injection of a standard solution of folate and amino acids (at 100 ng/mL) followed by injection of blank samples”

Unclear.

-Line 495-497. “For the analytes, the acceptance criterion for selectivity assessment was set at 20% of the peak area corresponding to the LLOQ level”

Unclear. Probably the authors mean that the measured area for potential interfering substances must be less than 20% of the measured areas for analytes at LLOQ?

Lines 510: “These reference materials were measured in triplicates in three runs. Similarly, the precision was considered suitable when the coefficient of variation of the replicates was lower than ± 15% for the QC samples”.

“Three runs” means that samples are injected three times or that samples are processed independently? In the first case (three replicate injections) the intra-day precision does not consider the variability introduced by sample preparation

-Formulas of recovery (I) and matrix effect  (II) are exactly the same: “(A-B)/Cx100”.

Something is wrong.

Moreover the meaning of A, B and C is not clear. For example, for the formula of matrix effect (II) it is explained that:

“C is the mean peak area of the QCs in the extraction solution” (lines 537-538)

It is well known that to calculate the matrix effect, the term in the denominator must be a peak area measured in a standard solution (without a matrix). So what means “extraction solution”?

-Table 1: results expressed with a standard deviation equal to “zero” (+/- 0.00) are clearly inconsistent because each measure is affected by an error.

-Table 4. The title “Table 4. Regression equations and limits of detection and quantitation of eleven compounds.” The title shoud be “Table 4. Linearity ranges, limits of detection and quantitation of the eleven compounds”. There are not “regression equations”

-As already stated, Figure 3 in not readable since there are too many chromatograms. 

Comments on the Quality of English Language

There are no major grammatical errors. The problem is that the concepts are not always expressed in fluent English.